# Field-Induced Assembly of sp-sp^2^ Carbon Sponges

**DOI:** 10.3390/nano11030763

**Published:** 2021-03-17

**Authors:** Stella Kutrovskaya, Vlad Samyshkin, Anastasia Lelekova, Alexey Povolotskiy, Anton Osipov, Sergey Arakelian, Alexey Vitalievich Kavokin, Alexey Kucherik

**Affiliations:** 1School of Science, Westlake University, 18 Shilongshan Road, Hangzhou 310024, China; a.kavokin@westlake.edu.cn; 2Institute of Natural Sciences, Westlake Institute for Advanced Study, 18 Shilongshan Road, Hangzhou 310024, China; 3Department of Physics and Applied Mathematics, Stoletov Vladimir State University, 87 Gorkii Street, 600000 Vladimir, Russia; samyshkin@vlsu.ru (V.S.); alelekova@vlsu.ru (A.L.); osipov@vlsu.ru (A.O.); arak@vlsu.ru (S.A.); kucherik@vlsu.ru (A.K.); 4Institute of Chemistry, Saint Petersburg State University (SPSU), Universitetskaya nab. 7/9, 199034 Saint Petersburg, Russia; alexey.povolotskiy@spbu.ru; 5ILIT RAS Branch of FSRC Crystallography and Photonics RAS, 140700 Shatura, Russia; 6Russian Quantum Center, Skolkovo IC, Bolshoy Bulvar 30, bld. 1, 121205 Moscow, Russia

**Keywords:** carbon allotropes, Raman characterization, photoluminescence spectra

## Abstract

The formation of macroscale carbon structures characterised by an sp-sp${}^{2}$-hybridization is realised by self-assembly in colloidal solutions under an effect of laser irradiation and electromagnetic fields. The sponge-like morphology, sculptured with gold nanoparticles (NPs) was revealed by Scanning Electron Microscopy (SEM) imaging. Full structural and defect characterization of the self-assembled sponges was provided using the micro-Raman spectroscopic technique. The synthesized clusters manifest themselves in the presence of a strong spectral band in the visible range of the photoluminescence spectra that is quite unusual for ordered sp${}^{2}$-carbon systems.

## 1. Introduction

Carbon is a widespread chemical element whose atoms may be linked by single, double or triple bonds with neighboring atoms. This variety of options for chemical bonds allows for the formation of a huge variety of possible carbon allotropes that demonstrate a rich variety of electronic and optical properties. The quantum confinement effect offers an efficient tool of control of these properties, which is why carbon-based nanostructures are currently in the focus of a significant interest. Since the beginning of the XXIst century, several low-dimensional carbon nanosystems have come into play, such as fullerenes, carbon nanotubes (CNT) and graphene. They are promising for applications in nanoelectronics, photonics, THz-science and so forth [1]. In this context, there is a clear technological need for the development of reliable and easy-to-use methods of synthesis of carbon-based nanostructures with tailored electronic and optical properties. An efficient control over the results of carbon synthesis may be provided by noble metal nanoparticles (NPs) serving as catalysers. As an example, the recent publication [2] demonstrated a convenient approach for fabrication of highly stretchable supercapacitors based on vertically aligned nanotubes and nanowires with embedded gold NPs that allowed achieving an exceptional and robust electrochemical performance. The graphene sheets with incorporated metal NPs [3] are highly promising for THz-generation because of the possibility to inject hot carriers from the NPs to the graphene matrix. Also, linear sp-carbon chains stabilized by gold anchors attached to their ends [4] exhibit the formation of edge electron-hole states that trigger formation of bright excitons and trions.

This work is aimed at the demonstration of self-assembly of sp-sp${}^{2}$ carbon sponges in liquid media. The metal NPs were used as donors of extra carriers in a solution as well as the growth points for the assembly of carbon matter. The near infrared (NIR) laser action was employed to speed up progression of chemical reactions in the colloidal media. To reduce multicompound charged complexes into simpler ordered carbon structures we used two electrodes immersed into the solution. The Raman and PL spectra reveal the singularities of the sp-sp${}^{2}$ carbon sponges.

## 2. Materials and Methods

We employ the electric-field assisted technique to achieve the formation of macroscale sponge-like structures based on sp-sp${}^{2}$ carbon crystals. The initial solution contains disordered shungite particles (their structure is addressed in detail in Section 3.2) distributed in the volume of distilled water (DI). Spherical Au NPs of about 20 nm diameter were used for the realization of the electric field driven transport of the colloid fraction bearing a permanent electric charge. The transmission electron microscopy (TEM) images of shingite target and gold NPs are given in the Appendix A. To achieve a laser activation of the colloidal system we have used nanosecond laser pulses generated by an Ytterbium (Yb) fiber laser having the central wave length of 1.06 m, the pulse duration of 100 ns, the repetition rate of 20 kHz and the average power of 20 W. The laser spot size was about 50 μm, however, the area of IDs interaction was governed by the interplay of laser scattering and heating. The laser beam was scanning a rectangular area without overlapping between layers that starts from the anticathode and leads to an opposite direction with a speed of 15 mm/s. The sizes of gold NPs and the zeta-potential of solutions have been controlled by the dynamic laser scattering device Horiba SZ100. The carbon phase configuration was investigated with the use of Raman Senterra spectrometer made by Bruker with excitation at 532 nm. The collection time was 10s, each spectra was a result of averaging over 9 measurements. All the studies of liquid solutions reported here were conducted with the use of fused quartz cuvettes. The homogeneous electric field was provided by two parallel plane electrodes separated by 3 cm from each other. The assembly of carbon sponges was performed in the presence of an electric field of 4 V/cm. Note that at any lower value of the applied field the DI water molecular dissociation would never start in our setup. For the detailed study of the synthesised complexes, we have performed the high resolution transmission electron microscopy and scanning electron microscopy SEM using FEI Titan${}^{3}$ with a spatial resolution of up to 2 Å and Quanta 200 3D with an energy-dispersive X-ray analyser (EDAX) column with a spatial resolution of up to 7 nm. To investigate optical properties of sponges, the dark sedimentation was deposited from a water solution on a substrate and dried at room temperature. We have used a silica glass as a substrate for the photoluminescense studies and a gold-platinum grid of the 400 mesh as a substrate for high-resolution transmission electron microscopy (TEM) measurements. The 10.0 ± 0.5 μm thickness of the obtained film placed on a glass was detected by the interference microscope. Then the absorbance spectra were measured by spectrophotometer PerkinElmer Lambda 1050 equipped with the 150 mm integrating sphere. The photoluminescence (PL) spectra were detected with a spectrometer Horiba Fluorolog-3, the excitation was made with an incoherent lamp light where the spectral line was selected by a dual monochromator with a spectral slit width of 5 nm. All the spectra were corrected accounting for the device’s hardware function.

## 3. Results

### 3.1. The Manifestation of a Self-Assembly Effect during the Carbon Growth

The electric-field assisted technique for formation of macroscale sponge-like structures of sp-sp${}^{2}$ carbon demonstrates on Figure 1.

Under the effect of near infrared (NIR) laser action, metal NPs acquire extra charges and become a fairway of carbon growth [5] once it’s out of the laser heating zone. The amorphous carbon compound transforms to textured sp${}^{2}$-allotropes creating a spongy metal-carbon matrix. The formation of sp${}^{2}$-allotropes is confirmed by the Raman analysis. Interactions of induced dipoles (IDs) in a liquid with each other and the effect of inhomogeneous electric field varying as a function of the concentration of the colloidal fraction and of the laser field amplitude are the main driving forces for the synthesis of carbon-based microstructures. We observe, in particular, field- orienting hexagonal close packed crystals with built-in sp-defects (see Figure 1a). The field-induced parameters of the critical media enable the thermal decomposition of water into hydrogen and oxygen, especially in the vicinity of electrodes. The changing pH value leads to the reduction of zeta-potential which results in lowering of the energy barrier between the interacting particles thus strengthening the van der Waals interaction in the system. The sequential movement of a laser beam outside the anode’s area locally perturbes the equilibrium of our colloidal system and stimulates the process of self-assembly that results in the formation of a metal-carbon sponge, as Figure 1b–d show. Even though the morphology of carbon sponges has been characterised by SEM and HR TEM, the detailed Raman analysis is needed to determine the structure of carbon bonds.

### 3.2. The Raman Characterization

The presence of sp-sp${}^{2}$ carbon allotropes can be easily identified by Raman spectroscopy. The Raman spectra of carbon materials show characteristic features in the 500–3000 cm${}^{-1}$ region, where D${}^{\prime }$${}^{\prime }$, D, G, D${}^{\prime }$ peaks can be identified (see Figure 2). D${}^{\prime }$${}^{\prime }$ band around 1100 cm${}^{-1}$ is caused by the presence of single electron C-C mode vibrations.

D and D${}^{\prime }$ peaks correspond to the breathing modes of sp${}^{2}$ carbon atoms that certify of the presence of defects in the sp${}^{2}$ crystal lattice. G peak situated at ∼520 cm${}^{-1}$ corresponds to the E2g phonon at the center of the Brillouin zone. A possible contribution to the G peak may arise from the sp${}^{2}$ stretch vibrations of olefinic or conjugated carbon chains at about 1600 cm${}^{-1}$ [6]. Defects induce a significant increase of the intensities of D and D${}^{\prime }$ bands and the associated decrease of the G-peak that is characteristic of a highly oriented crystalline phase.

The Raman spectrum of the initial precursor carbon system is shown in the top panel of Figure 2. The major spectral features, namely D${}^{\prime }$${}^{\prime }$, D, G and D${}^{\prime }$ bands appearing at 1100, 1330, 1518 and 1632 cm${}^{-1}$, respectively, are characteristic of sp${}^{2}$ carbon. Low-frequency Raman bands of 754, 825 and 910 cm${}^{-1}$ are related to the sp${}^{2}$-phase activated by disorder. In particular, they may be assigned to the in-plane rotation of sixfold carbon rings. The full spectrum composition is indicative of the presence of only lattices of sp${}^{2}$-carbon with a high defect concentration in the initial solution. In order to characterize the structural properties of the system after the field-induced carbon transformation we have investigated both substances: the assembled carbon sponge and the depleted colloidal media (see the bottom panel in Figure 2). The evolution of carbon allotropes is also apparent in the modification of the Raman spectra. Here one can see the pronounced G and D peaks. The existence of LLCC band at about 2100 cm${}^{-1}$ is typically associated with a stretching mode of sp-hybridized carbon. A peak centered between D and G - modes, namely at 1462 cm${}^{-1}$, can be attributed to the semicircle stretching of carbon atoms in finite-size crystals of graphite. The second order of zone boundary phonons gives rise to the 2D peak. In the most part of cases it appears either as a strong single peak (in graphene) or a broadened band (in graphite). We attribute the presence of a narrow peak at 815 cm${}^{-1}$ to the formation of single electron bonds between Au NPs and carbon matrix. It’s noticeable that the lengths of this type of bonds exceed those of single C-C pairs.

### 3.3. Photoluminescence and Absorbance Study

Typical sp${}^{2}$-carbon crystals, such as graphite or graphene are semimetals that do not emit visible light. Moreover, they are generally classified to be strong wide spectral range absorbers. Carbon nanotubes are also poor light emitters because of the dark exciton effect [7]. Moreover, the luminescence in carbon nanoparticles is not caused by the carbon core, but it is dominated by the surface endcapped groups, structural defects and edge carbon atoms [8]. On the other hand, sp-carbon crystals demonstrate a remarkable ability to emit visible light whose wavelength varies as a function of the chain length. He we propose our system of sp-sp${}^{2}$ carbon allotropes for different applications in photonics: we have found a good PL response of the depleted colloidal media from Figure 2. and high absorbance ability for a sedimented sponge. In order to reveal their potentiality for emission (see Figure 3a) and absorption (Figure 3b) of light we use a state-of-art inverse band structure design approach. Due to structural changes in molecular orbitals on a length-scale of the carbon chain, the emission properties of sp-carbon are also highly sensitive to the excitation wavelength. The band gap of finite size linear chains dramatically depends on the number of carbon atoms in a chain, getting larger in shorter chains. Here we have studied the integrated PL response from sp-sp${}^{2}$ sponges in the spectral range of 330–418 nm (see Figure 3a) that corresponds to the band of strong absorption of light in straight linear chains containing from 8 to 24 carbon atoms [4]. If the same system is excited at the same energy as linear chains that we studied previously, logically, the variation of the position of its PL maximum with respect to the emission peaks of linear chains would be governed by the magnitude of the Stokes shift characteristic of our structure. Simultaneously, the line intensity depends on the efficiency of excitation of HOMO-LUMO electron transitions in sp-linear chains. The Stokes shift dependence on the excitation wavelength is monotonous, as the inset to Figure 3a shows. One also can see that the absorption spectrum of the deposited sponge in Figure 3b demonstrates a huge absorption coefficient in VIS-NIR spectral ranges. We conclude that the obtained materials based on carbon sponges are highly promising for application in antireflection coatings efficient in a wide and most relevant spectral range.

## 4. Discussion

Let us now compare the microscopy data shown in Figure 1. with the Raman data presented in Figure 2. There are three major parameters to control sp-sp${}^{2}$ growth in our setup, namely, electric and laser field strengths, and the concentration of disperse fraction. The latter one determines the interacting IDs. First of all, let us briefly discuss the role of electric and laser fields. We assume the Van der Waals interactions for IDs are negligible in the absence of the fields. This is because the neutral carbon constituent is surrounded by the liquid media and negatively charged Au NPs in this case. Metal NPs formed by laser ablation method in liquids are typically encircled by a double containment layer (DCL) [9]. The medium laser action leads to the decomposition of the amorphous carbon compound that is certified by a full disappearance of the D${}^{\prime }$ peak in Figure 2 and the transformations of G and D+G peaks into the bands characteristic of linear sp-chains and into shorter elements including even free atoms in the beam area. The presence of these elements enables formation of textured sp${}^{2}$-allotropes outside the laser heating zone.

Laser irradiation also activates electrons in metal NPs that supply extra charges to carbon nanostructures and locally increases temperature of a some solution volume under a laser beam, then it pushes in active diffusion of counter-ions and DCL expansion. The mechanisms of ID-ID interaction include the electrostatic interaction due to static surface charges and the dipole-dipole interaction due to applied electric fields. We emphasize the importance of field-induced dissociation of DI water molecules that in the vicinity of anti-cathode location leads to the significant reduction of the pH-factor [10]. This is also confirmed by our zeta-potential measurements. The reduced zeta potentials trigger the reduction of the activation barrier characterising the transition to the regime where the Van der Waals force dominates over other interaction mechanisms [11,12]. In our case, Van der Waals interactions become essential locally in the anti-cathode vicinity where the sponge growth starts from. At that stage the total carbon concentration in the colloid serves as a factor of control of the full carbon composition forming in the vicinity of the anti-cathode: at the concentration of about 100 mg/ml we obtain an sp${}^{2}$ lattice doped with the sp-fragments, while at the concentration of 1000 times less one can obtain the sp-linear carbon chains doped with sp${}^{2}$ elements. A metastable sp-constituent is formed preferentially at the high-temperature and low-density conditions, while the sp-sp2 transformation is achieved in the high-density regime, as it was found in the case of the nanostructure growth in a carbon plasma [13].

## 5. Conclusions

In conclusion, we have developed an experimental setup based on the field-induced formation of tailored carbon-based sponges. We have synthesized sp-sp${}^{2}$ sponges of several mm sizes under the movable laser beam action at the homogeneous electric field. The field-induced assembly offers a powerful tool for the carbon phase separation for a deposited carbon sponge and carbon-containing liquid media, as confirmed in this work by means of the Raman analysis. The designed hybrid carbon allotrope of mostly sp-hybridisation demonstrates a stable PL response in the visible range that makes it promising for applications in photonics and THz detectors. On the other hand, the sponge-like matter is highly suitable for antireflection applications in the VIS-NIR ranges because of its huge absorption coefficient.

## Figures and Tables

**Figure 1 nanomaterials-11-00763-f001:**
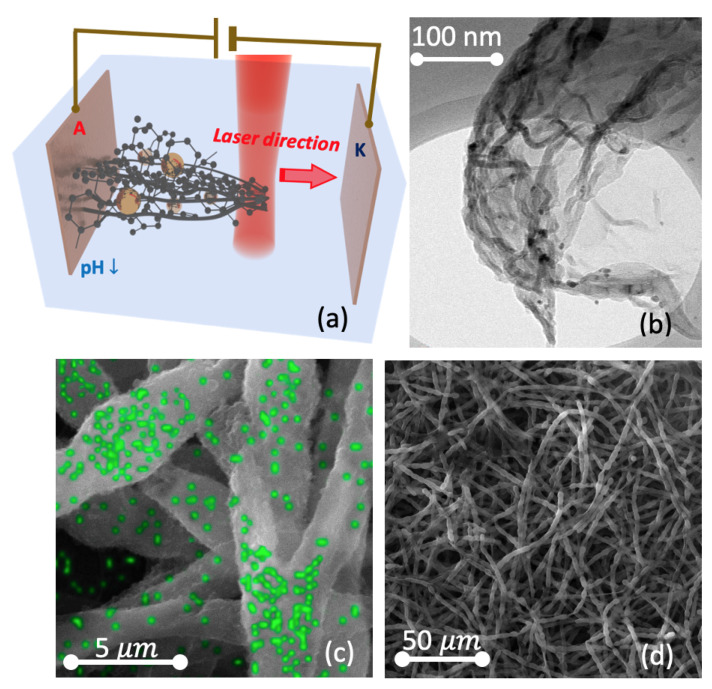
The top panel (**a**) illustrates the concept of self-assembly of sponge-like structures based on sp${}^{2}$ carbon doped with a high concentration of sp-carbon bonds. The carbon system is injected by auxiliary charges due the vicinity of endcapped Au NPs (shown as yellow spheres). Near the anti-cathode the pH-factor of the DI colloidal system falls down giving start to the sponge assembly. The elementary part of a peeled sponge has been discovered by the high resolution transmission electron microscopy (HR TEM) on (**b**), where dark points correspond to the Au NPs. The full picture at a macroscale is shown in panels (**c**,**d**). These images reveal the bunch arrangement composing the sponge body of at least few mm in its size. The panel (**c**) combines the scanning electron microscopy (SEM) image and with the results of EDAX element analysis in order to make the Au NPs accumulation visible (gold distribution marked by green spots). One can see gold nanoparticles (Au NPs) accumulated in the centers of each bunch segments and splitters. This shows the role of NPs as growth points for the carbon assembly.

**Figure 2 nanomaterials-11-00763-f002:**
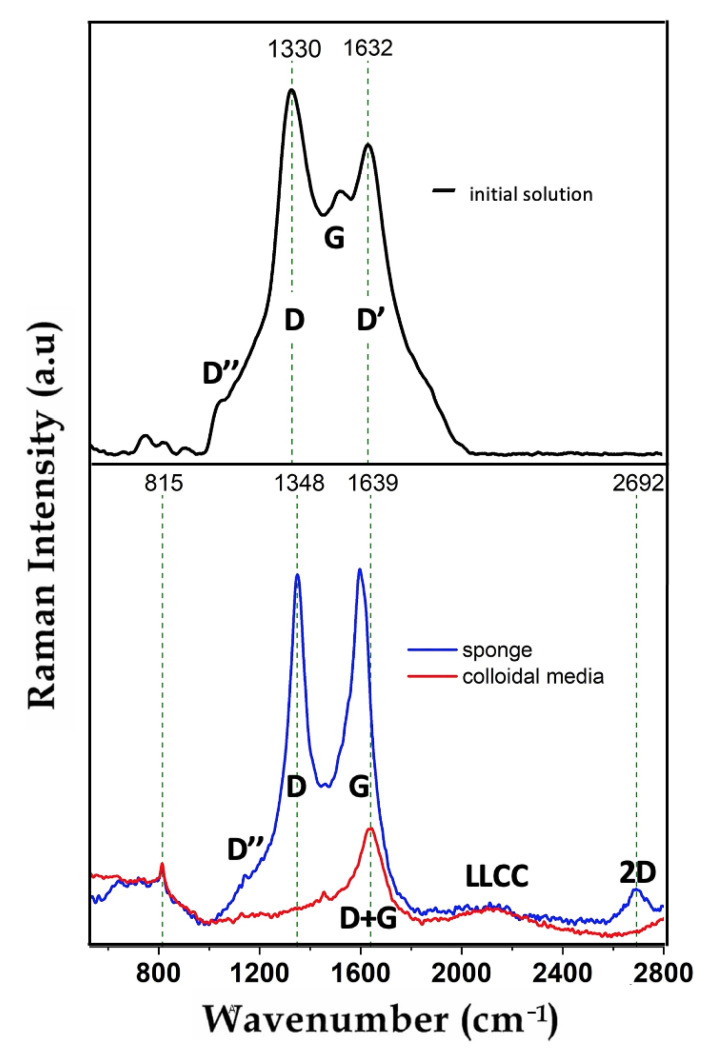
Raman spectra: the top panel is relevant to the initial colloidal system, the bottom panel shows the Raman spectrum of the system after the field-induced assembly. The spectra were taken separately for a sedimented sponge (marked by the blue curve) and the depleted colloidal media (marked by red curve).

**Figure 3 nanomaterials-11-00763-f003:**
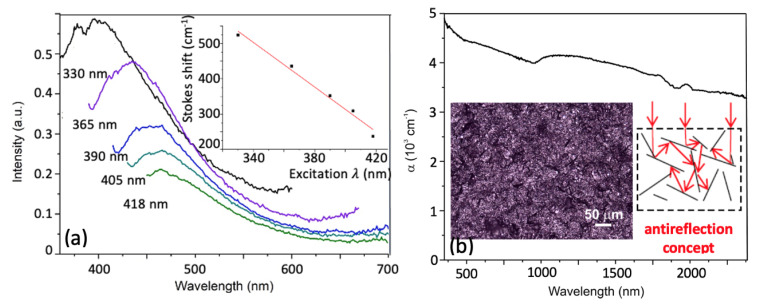
(**a**) The photoluminescence spectra of a depleted liquid phase of the studied carbon system registered at different excitation wavelengths. The inset shows the Stokes shift dependence on the excitation wavelength; (**b**) the absorption spectrum of a deposited sponge film whose optical image is shown in the inset. The concept of increased light absorption due to high porous sponge surface is schematically illustrated on the right hand side of the insert of the Figure 3b.

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
