# Peer review of "Field-Induced Assembly of sp-sp2 Carbon Sponges"

_nanomaterials, 2021, doi:10.3390/nano11030763_

Round 1
Reviewer 1 Report
The topic of carbon based nanostructures is of current interest and the manuscript under review presents a timely report on the synthesis and characterization of macroscale sp-sp2 carbon structures decorated with gold nanoparticles. The carbon materials were formed by self-assembly in solutions using of laser irradiation and electromagnetic fields and characterized by SEM and optical spectroscopies (Raman, photoluminescence, and absorbance). The work contains some interesting results but the analysis and too brief and lack necessary explanations.
Experimental particulars regarding the actual preparation of the carbon structures (starting materials, pH, Au particle incorporation…) are very limited as are the sample preparations for the optical measurements. These need to be outlined more fully.
In several places in the Results and the Discussion sections the explanations are rather terse. For example the authors state ‘ Moreover, the luminescence in carbon nanoparticles isn’t caused by the carbon core, but it is dominated by the surface endcapped groups, structural defects and edge carbon atoms[8].’ Giving a reference is insufficient, this needs a more detailed explanation.
In another place the authors claim ‘….the integrated PL response from sp-sp2 sponges in the spectral range of 330-418nm (see Figure 3a) that corresponds to the band of strong absorption of light in straight linear chains containing from 8 to 24 carbon atoms [4].’ Again, just a reference. As a reader, I want to see an explanation that follows.
Did UV-vis spectra from different carbon sponges samples look similar? Did film thickness play any role in the spectral appearance?There is no explanation how the Au particles influenced the PL and UV-vis optical spectra.
Have the authors taken any XPS data of their carbon sponges?
Author Response
Reviewer 1 (Comments and Suggestions for Authors):
The topic of carbon based nanostructures is of current interest and the manuscript under review presents a timely report on the synthesis and characterization of macroscale sp-sp2 carbon structures decorated with gold nanoparticles. The carbon materials were formed by self-assembly in solutions using of laser irradiation and electromagnetic fields and characterized by SEM and optical spectroscopies (Raman, photoluminescence, and absorbance). The work contains some interesting results but the analysis and too brief and lack necessary explanations.
Authors: we appreciate this Reviewer’s assessment. The required corrections have been added to the manuscript according to the Reviewers’ suggestion, all new parts have been coloured for the readers convenience.
Reviewer 1: Experimental particulars regarding the actual preparation of the carbon structures (starting materials, pH, Au particle incorporation…) are very limited as are the sample preparations for the optical measurements. These need to be outlined more fully.
Authors: In the new version of the manuscript, we have added the section entitled “The shungite precursor and gold nanoparticles” into the Appendix A that fully addresses this reviewer’s comment. The section of Materials and Methods now also contains the details on the sample preparation for optical measurements.
Reviewer 1: In several places in the Results and the Discussion sections the explanations are rather terse. For example the authors state ‘ Moreover, the luminescence in carbon nanoparticles isn’t caused by the carbon core, but it is dominated by the surface endcapped groups, structural defects and edge carbon atoms[8].’ Giving a reference is insufficient, this needs a more detailed explanation.
Authors: We would like to emphasize at this point that the discussion of sp2-carbon emission starts in the earlier section 3.3 Photoluminescence and absorbance study. There we mentioned that the well-known sp2 carbon family representatives, such as graphite, graphene, sp2-quantum dots (QD) are semimetals. Their electronic band structure is characterised by valence and conduction bands forming conical surfaces in the momentum space that are touching at Dirac Points in the particular case of graphene. There is no optical band gap in these materials. A consequence of this lack of a bandgap is that pure sp2 crystals cannot be efficiently characterised by means of the photoluminescence (PL) spectroscopy: as a rule, zero bandgap crystals do not produce resonant PL features. However, by oxidising graphene to form graphene oxide, the π‑electron network of the carbon atoms in graphene is disrupted which results in the formation of a gap between the valence and conduction bands. In this regime, the material becomes photoluminescent, which opens the door to its applications in optoelectronics. The same is valid for sp2 - QD with surface endcapped groups, structural defects or edge carbon atoms.
Reviewer 1: In another place the authors claim ‘….the integrated PL response from sp-sp2 sponges in the spectral range of 330-418nm (see Figure 3a) that corresponds to the band of strong absorption of light in straight linear chains containing from 8 to 24 carbon atoms [4].’ Again, just a reference. As a reader, I want to see an explanation that follows.
Authors: Accounting for this comment, we have added a supplementary discussion to the manuscript now (row 143): “The band gap of finite size linear chains dramatically depends on the number of carbon atoms in a chain, getting larger in shorter chains.” The dependence of absorption peaks in linear chains on the number of atoms composing a chain has been studied in several publications, e.g. [Shi, L.; Rohringer, P.; Suenaga, K.; Niimi, Y.; Kotakoski, J.; Meyer, J. C.; Peterlik, H.; Wanko, M.; Cahangirov, S.; Rubio, A.; Lapin, Z.; Novotny, L.; Ayala, P.; Pichler, T. Conned linear carbon chains as a route to bulk carbyne. Nat. Mater. 2016, 15, 634−639.; Casari, C. S.; Tomassini, M.; Tykwinski, R. R.; Milani, A. Carbon-atom wires: 1-D systems with tunable properties. Nanoscale 2016, 8, 4414.]
Reviewer 1: Did UV-vis spectra from different carbon sponges samples look similar? Did film thickness play any role in the spectral appearance? There is no explanation how the Au particles influenced the PL and UV-vis optical spectra.
Authors: for the studied sponges that have been deposited in a liquid phase on a substrate and dried out to form a film of 10.0±0.5 thickness the UV-VIS spectra remain qualitatively the same. We have studied the integrated in space and stable in time optical response of a macroscopic sponge. We did not study nonlinear effects that might play an important role on a nanoscale here. We expect the appearance of new spectral features and peculiar non-linear effects in the films of thicknesses less that , which is beyond the scope of our present study.
Taken alone, gold nanoparticles of 20nm do not exhibit any photoluminescence. However, being combined with linear carbon chains, they may form elongated quasicrystals based on monoatomic polyyne wires attached to a gold NPs. This quasicrystal does emit light indeed. The PL data on it are discussed in detail in [S. Kutrovskaya, et al, Nanoletters , 20(9), 6502-6509 (2020).]. The role of Au NPs in this particular case is twofold: they are responsible for the linear chain stabilization and for the exciton localization at the ends of the carbon chains that results in an equivalent of Su-Schrieffer-Heeger states. Our ab-initio calculations confirme that HOMO and LUMO electron densities would be spread over the whole chain in the absence of gold anchors at the ends.
It should be added that Au NPs give strong resonant contributions to the absorption spectra where the plasmon line may be found in the spectral range of 510-550 nm. Its exact position depends on the Au NP diameter.
Reviewer 1: Have the authors taken any XPS data of their carbon sponges?
Authors: Unfortunately, we could not use the XPS-analysis in the frame of this work.
Reviewer 2 Report
In this paper the authors described the preparation of macroscale carbon structures, characterised by an sp-sp2 hybridization, using an electric-field assisted laser ablation technique starting from a shungite colloidal suspension. Gold nanoparticles were added to the shungite colloid with the dual purpose of catalyzing the reaction and acting as a support for the carbon structure. The obtained materials were characterized by electron microscopy, Raman spectroscopy and photoluminescence spectroscopy.
Graphyne carbon allotropes have interesting properties, and their synthesis is challenging, consequently this article is potentially interesting. Nevertheless, there are a few questions and suggestions I would like the authors to address.
- The synthetic procedures employed for the preparation of these materials are not explained clearly enough. In particular:
- I believe that the starting shungite colloidal suspension was produced by the authors, but details of its synthesis and characterization are missing. It would also be interesting to include SEM micrograph of these particles.
-What is the concentration of carbon and gold nanoparticles employed? In the Discussion section (page 6 row 178) the authors state that the concentration of the carbon particles influences the obtained carbon composition, there is any influence of the gold nanoparticles concentration or carbon-gold ratio?
-What is the total irradiation time? There is any influence of the irradiation time on the obtained carbon structure?
- The author recently published a paper where, with an apparently very similar procedure they obtained very different carbon structures. This paper should be commented in the discussion.
- The authors claim that the formation of the carbon structure is obtained through a self-assembly process but the structural properties of the system change significantly during the electric-field assisted laser ablation process so I don't believe this term can be employed.
Author Response
Reviewer 2 Comments and Suggestions for Authors
In this paper the authors described the preparation of macroscale carbon structures, characterised by an sp-sp2 hybridization, using an electric-field assisted laser ablation technique starting from a shungite colloidal suspension. Gold nanoparticles were added to the shungite colloid with the dual purpose of catalyzing the reaction and acting as a support for the carbon structure. The obtained materials were characterized by electron microscopy, Raman spectroscopy and photoluminescence spectroscopy.
Graphyne carbon allotropes have interesting properties, and their synthesis is challenging, consequently this article is potentially interesting.
Authors: we appreciate this Reviewer assessment.
Reviewer 2: Nevertheless, there are a few questions and suggestions I would like the authors to address.
The synthetic procedures employed for the preparation of these materials are not explained clearly enough. In particular:
- I believe that the starting shungite colloidal suspension was produced by the authors, but details of its synthesis and characterization are missing. It would also be interesting to include SEM micrograph of these particles.
Authors: We have added the auxiliary details and TEM images of the starting materials in the Appendix A. We are confident that in the present form it fully meets the reviewer’s suggestion.
Reviewer 2: -What is the concentration of carbon and gold nanoparticles employed? In the Discussion section (page 6 row 178) the authors state that the concentration of the carbon particles influences the obtained carbon composition, there is any influence of the gold nanoparticles concentration or carbon-gold ratio?
Authors: In the amended version of the manuscript, in the Appendix A we explicitly mention the starting materials concentrations, being C:Au:H2O 10-2:10-4:1. In our fabrication setup metal NPs were used mostly as donors of extra carriers in a solution that give rise to carbon growth points. The specific chosen composition ratio is determined by the fact that the free surface of nanoparticles is large enough to accommodate many single-electron bonds linking carbon structures to every gold NP. Due to these multiple bonds gold NPs find themselves incorporated in a sponge structure. The further increase of the gold concentration leads only to the gold cluster agglomeration which is not addressed in this study.
Reviewer 2: -What is the total irradiation time? There is any influence of the irradiation time on the obtained carbon structure?
Authors: In our setup the laser beam was employed in a scanning regime on a rectangular area that started from the anticathode. The beam was moved with a speed of 15mm/s. The irradiation time was dependent on a targeted size of the resulting structure. For example, it took us 30 mins to form a 4 mm long sponge.
Reviewer 2: The author recently published a paper where, with an apparently very similar procedure they obtained very different carbon structures. This paper should be commented in the discussion.
Authors: Indeed, during 2 last years we have published a set of works aimed at the synthesis, stabilization and optical study of long sp-carbon chains. The method employed in the present work is similar to the methods we used in the previous works in what concerns the used precursors (shungite and nobel metals), 1 mkm laser light and the liquid media. However, there are several important differences between the approaches used in the previous works and here. In particular, here the carbon concentration in the growth area is 3-5 orders of magnitude higher than in our previous works. Also, in contrast to the previous studies, here we use an external electric field that triggers the dipole-dipole interactions and accelerates chemical reaction in the solution. In these conditions a dominated linear carbon chain growth is unlikely, and even if some percentage of linear sp-chains would be forming, they would tend to undergo the sp^2 transition while being in the active heating zone out of direct laser beam. The resulting structures in our previous works were atomically thin films formed by single carbon chains or their bunches. In the present study we fabricate carbon sponges of a macroscopic thickness. In the present version of the manuscript we clearly distinguish between methods employed in this and previous works.
Reviewer 2: The authors claim that the formation of the carbon structure is obtained through a self-assembly process but the structural properties of the system change significantly during the electric-field assisted laser ablation process so I don't believe this term can be employed.
Authors: Actually, we study the formation of sponges from C-Au fractions in DI solution due to the self-assembly process under the stimulating of external effects of laser and electric fields. We reckon that the resulting structures are as self-assembled, but not self-organized. This is because only the fragments of the original structure form the resulting complex structure, there is no evidence of any self-organization whatsoever. Moreover, we would like to emphasize that the formation of sponges has never been observed in the absence of the external fields. The assembly process was triggered externally that excludes self-organization, as far as we could see.
Round 2
Reviewer 1 Report
Responses to reviewer's comments are satisfactory and the manuscript is acceptable for publication.
Author Response
We appreciate this Reviewer’s assessment
Reviewer 2 Report
I checked the revised version of the paper and the authors' answers and I believe that the paper is now suitable for publication in Nanomaterials, there are only a couple of typos to correct:
1. pag2 row 45: In figure 1 are reported TEM images;
2. Fig. A1 b) the scale bar should be 10 nm.
Author Response
We thank the Reviewer for the careful reading of our manuscript, useful comments and the positive overall assessment. In the resubmitted version of the manuscript we have fully taken into account the Reviewer remarks.